# Observation of long-range tertiary interactions during ligand binding by the TPP riboswitch aptamer

**Van K Duesterberg[1†], Irena T Fischer-Hwang[2†], Christian F Perez[3†], Daniel W Hogan[4], Steven M Block[4,5]\***

[1]Biophysics Program, Stanford University, Stanford, United States; [2]Department of Electrical Engineering, Stanford University, Stanford, United States; [3]Department of Physics, Stanford University, Stanford, United States; [4]Department of Applied Physics, Stanford University, Stanford, United States; [5]Department of Biology, Stanford University, Stanford, United States

**Abstract** The thiamine pyrophosphate (TPP) riboswitch is a *cis*-regulatory element in mRNA that modifies gene expression in response to TPP concentration. Its specificity is dependent upon conformational changes that take place within its aptamer domain. Here, the role of tertiary interactions in ligand binding was studied at the single-molecule level by combined force spectroscopy and Förster resonance energy transfer (smFRET), using an optical trap equipped for simultaneous smFRET. The 'Force-FRET' approach directly probes secondary and tertiary structural changes during folding, including events associated with binding. Concurrent transitions observed in smFRET signals and RNA extension revealed differences in helix-arm orientation between two previously-identified ligand-binding states that had been undetectable by spectroscopy alone. Our results show that the weaker binding state is able to bind to TPP, but is unable to form a tertiary docking interaction that completes the binding process. Long-range tertiary interactions stabilize global riboswitch structure and confer increased ligand specificity.

**\*For correspondence:** sblock@stanford.edu

[†]These authors contributed equally to this work

**Competing interests:** The authors declare that no competing interests exist.

## Introduction

The gene-regulatory activity of a riboswitch is mediated by its ability to form a substructure, called the aptamer domain, that binds—and thereby senses—a ligand, which is generally a small metabolite (*Roth and Breaker, 2009*; *Serganov and Nudler, 2013*; *Serganov and Patel, 2012*). Riboswitch aptamers fold and then bind their cognate ligands in a highly specific manner, and as they do so, they compete with alternative RNA structures that can form in conjunction with other domains of the riboswitch, which function either to permit, or to prevent, downstream gene expression. The aptamer of the TPP riboswitch senses the abundance of the coenzyme thiamine pyrophosphate, and in response, reduces the uptake of the vitamin thiamine. Versions of the TPP riboswitch have been discovered in all three kingdoms of life (*Cheah et al., 2007*; *Sudarsan et al., 2003*; *Winkler et al., 2002*). In *Arabidopsis thaliana*, the TPP aptamer is located upstream of the 3′ untranslated intron region of the *thiC* gene, and it contains a complementary sequence that, in the absence of TPP, acts to sequester a downstream splice site: it therefore functions by modulating alternative gene splicing (*Wachter et al., 2007*).

Crystal studies have identified striking similarities between the structures of the ligand-bound forms of a truncated TPP aptamer from *A. thaliana* and the *thiM* aptamer from *Escherichia coli* (*Edwards and Ferre-D'Amare, 2006*; *Serganov et al., 2006*; *Thore et al., 2006*). Specifically, both eukaryotic and prokaryotic aptamers share a common binding conformation (*Figure 1*),

**eLife digest** When a gene is switched on, its DNA is first copied to make a molecule of messenger ribonucleic acid (mRNA). The genetic code in the mRNA is then translated into a protein. There are also untranslated regions within mRNAs that do not code for protein themselves, but serve to regulate whether or not a protein is produced from the rest of the mRNA. For example, many mRNAs contain a motif in their untranslated region called a 'riboswitch'. These motifs selectively bind to molecules that are the products of metabolic processes. One riboswitch found in bacteria, animals and plants binds to a molecule known as thiamine pyrophosphate (TPP) and regulates genes that control the uptake of a vitamin called thiamine into cells.

Newly made mRNA molecules are linear strands that then fold into three-dimensional structures. The TPP riboswitch can adopt distinct shapes depending on whether it is bound to TPP or not. Knowledge of these structures is crucial for understanding how riboswitches regulate protein production. Previous research reported the folding of a TPP riboswitch from bacteria.

Here, Duesterberg et al. used a combination of two techniques known as force spectroscopy and Förster resonance energy transfer (FRET) to study the folding of the TPP riboswitch from a plant called *Arabidopsis thaliana*. The experiments show that in the presence of TPP, structural changes occur in two arm-like appendages – known as helix arms – that extend out of the core of the riboswitch. The riboswitch adopts a particular shape when TPP is strongly bound to it, and in this shape the riboswitch can regulate the activity of certain genes. However, if the riboswitch is only weakly associated with TPP, it takes on a shape in which the two helix arms are further apart and the riboswitch is unable to form the interactions required to complete the process of binding to TPP.

Duesterberg et al.'s findings reveal that the way in which the *A. thaliana* riboswitch changes shape when it is bound to TPP is different to that of its bacterial counterpart. The next challenge will be to observe these shape changes in even more detail, and to use these new techniques to study other riboswitches in various organisms.

characterized by contacts formed by (1) the TPP pyrimidine ring and the 'sensor' helix arm P2/3, (2) the TPP pyrophosphate and 'sensor' helix arm P4/5, and (3) a long-range tertiary rearrangement that brings together, and stabilizes, an interaction between the two arms carrying the L5 loop and the P3 stem (*Noeske et al., 2006*).

Results from single-molecule force spectroscopy experiments have suggested that TPP binding is modeled kinetically by two sequential states: initial weak binding, followed by strong binding (*Anthony et al., 2012*),

$$UF \longrightarrow F \underset{-TPP}{\overset{+TPP}{\rightleftharpoons}} F' \cdot TPP \rightleftharpoons F'' \cdot TPP \tag{1}$$

where UF refers to the unfolded form, and F, F' and F'' refer to folded forms. The existence of the folded states was inferred from measurements of the molecular extension of the aptamer as a function of the external force applied (force-extension curves, or FECs) during mechanically-induced unfolding. FECs obtained for the binding of TPP in the weakly bound, intermediate F' state were strikingly similar to FECs measured for the complete binding of two TPP analogs, thiamine monophosphate (TMP) or thiamine (T). These analogs lack one or both phosphate groups, respectively, resulting in incomplete binding to the bipartite ligand-binding site, which normally forms contacts with both the thiamine and phosphate moieties. Similar FECs were also measured for TPP binding by a mutant aptamer (*Anthony et al., 2012*) that is unable to form tertiary contacts between the two arms carrying L5 and P3. Taken together, these observations are consistent with two nonexclusive possibilities for the structure of the F'•TPP intermediate: (1) TPP may be incompletely coordinated at the binding site in this aptamer form, or (2) the aptamer structure may not have brought together its L5 and P3 arms, whose docking may be required for strong binding and the transition to the F''•TPP state.

Here, we address the structural ambiguity posed by the F'•TPP state by simultaneously monitoring fluorescent probes attached to the arms carrying L5 and P3, and aptamer folding, using an optical trap augmented with smFRET capabilities (*Figure 2*). By studying folding in the presence of TPP

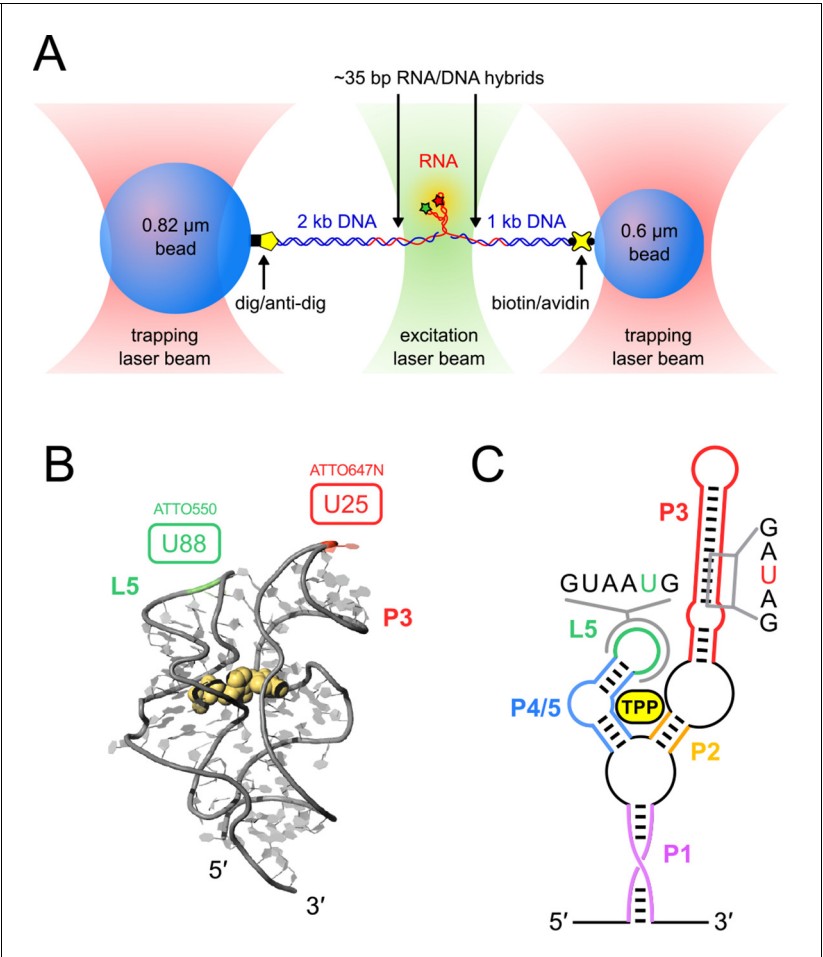

**Figure 1.** Single-molecule assay, crystal structure and schematic of the TPP aptamer. (**A**) Experimental geometry of the dumbbell optical-trapping assay with simultaneous FRET monitoring, with key components labeled (not to scale). (**B**) Crystal structure of the truncated TPP riboswitch aptamer with a shortened P3 stem (PDB entry 3D2G). Dye-labeling sites for ATTO550 and ATTO647N are indicated, at nucleotides G88 (green) and U25 (red), respectively. (**C**) Secondary structure of the TPP riboswitch, with structural components indicated in color. Sequences surrounding the amino-modified uracil bases in the sensor helix arms (5-N-U and 5-LC-N-U; Integrated DNA Technologies, Coralville, IA) are shown, with the modified bases colored (P3, red; L5, green).

and its analogs, we can explore the roles of long-range tertiary rearrangements and phosphate binding in ligand recognition.

Instruments that combine the capabilities of single-molecule fluorescence and optical trapping represent a comparatively new development (*Candelli et al., 2011*; *Comstock et al., 2011*; *Harada et al., 1999*; *Lang et al., 2004*; *Mameren et al., 2006*; *Sirinakis et al., 2012*; *van Mameren et al., 2009*), and these continue to present significant technical challenges (*Lang et al., 2004*; *van Mameren et al., 2008*). Recently, novel findings have begun to emerge using optical trapping and smFRET (*Comstock et al., 2015*; *Ngo et al., 2015*; *Suksombat et al., 2015*).

## Results

### Direct correspondence between P3-L5 configuration and secondary structural conformation

Using a dual-beam optical trap, we applied controlled loads to each end of an individual TPP aptamer RNA (3′ and 5′), and measured the end-to-end extension during mechanical unfolding (*Figure 1A*). Tertiary interactions were simultaneously scored by a FRET readout between donor and

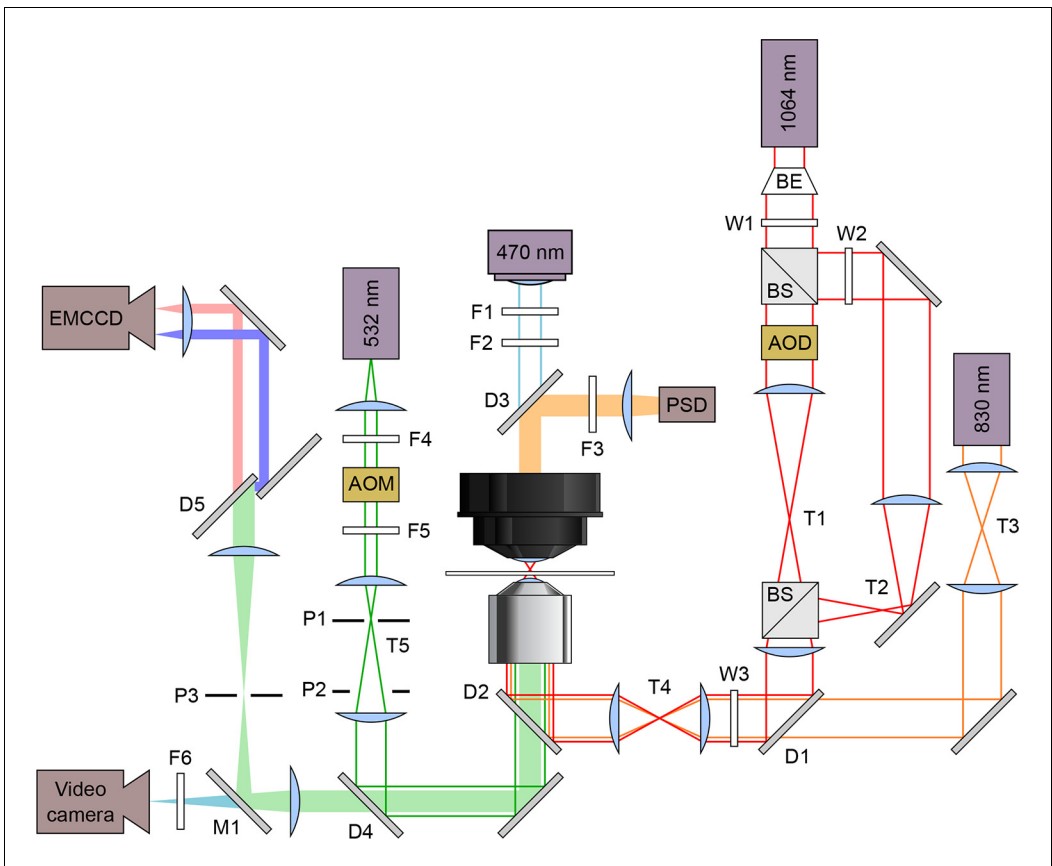

**Figure 2.** Schematic optical layout of the dual-beam optical trap and FRET. Solid lines indicate lasers and light sources: 830 nm detection laser (orange), 1,064 nm trapping laser (red), 532 nm excitation laser (green), and 470 nm illuminating LED (blue). Filled bars are emissions received by these detectors: position sensitive detector (PSD), electron multiplying charge coupled device (EMCCD) camera, and a video camera. Lasers are maneuvered and modified in intensity using acousto-optic deflectors (AOD) and acousto-optic modulators (AOM), respectively. The dual beam trapping laser is expanded using a beam expander (BE) and split with a beam splitter. The excitation laser emission goes through a dichroic mirror (D5, T640LP; Chroma, Brattleboro, VT) to split the acceptor and donor emissions. Filters (F), Wollaston prisms (W), pinholes (P), dichroic mirrors (D), mirrors (M), and telescopes (T) are labeled.

acceptor fluorescent dyes, ATTO550 and ATTO647N (*Figure 1B*), which were covalently attached to the L5 and P3 helix arms, respectively. Aptamers were subjected to repeated cycles of unfolding and refolding. After a preliminary unfolding of the RNA, the optical traps were brought into close proximity, which reduced the applied load below 5.0 pN for a time of 0.5–30 s, allowing the aptamer to refold under low force (the refolding step). The force was subsequently ramped up by increasing the trap separation linearly at 200 nm/s (*Figure 3*, *Figure 3—figure supplement 1*, *2*), as both molecular extension and fluorescence signals were acquired. After full extension, the refolding/ unfolding cycle was repeated, until either a dye photobleached or the RNA tether broke. As reported previously, FECs could be categorized into one of three distinct forms (*Figures 3A,B,C*) (*Anthony et al., 2012*), each corresponding to a different folded conformation, as follows. In the absence of TPP, FECs for the folded aptamer (F) displayed three opening transitions, or rips, corresponding to the sequential rupture of the secondary structural elements (P1 + P2), P4/5 and the P3 stem, respectively. In the presence of TPP, two additional FEC forms were observed: (1) an FEC with a single opening transition at low force (<20 pN), sometimes followed by an additional opening transition, that corresponds to the weakly bound state, F'•TPP, and (2) an FEC with a single, high-force (>20 pN) transition, that corresponds to the tightly bound, fully folded state, F"•TPP. FRET signals acquired during force ramps could be categorized into three groups, with characteristics that were attributable to each of the three established aptamer conformations, F, F'•TPP, and F"•TPP. In the absence of TPP (*Figure 3A*), the FRET efficiency of the F state remained stable near a low value

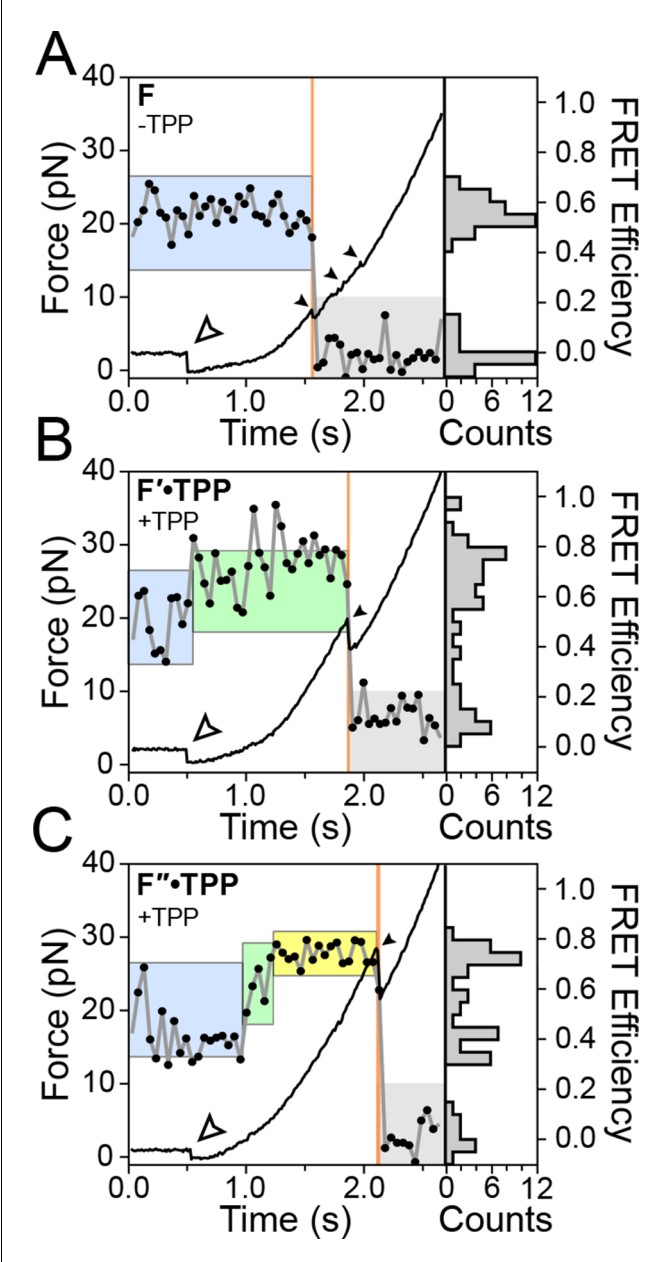

**Figure 3.** Representative FEC and FRET traces. Representative traces of unfolding for the aptamer conformations (**A**) F, (**B**) F'•TPP, and (**C**) F''•TPP. Simultaneous force-extension curves (FECs; black lines) and FRET trajectories (black circles, gray lines) are parametrized by time. Colored boxes indicate the APO (blue), WB (green), and SB (yellow) FRET states. Open arrowheads mark the end of the refolding period and the start of the force ramp. Small, filled arrowheads mark the location of opening transitions.

The following figure supplements are available for figure 3:

**Figure supplement 1.** Representative traces of various secondary state conformations and TPP concentrations.

**Figure supplement 2.** FRET changes occur throughout the refolding period and the force ramp, up until the first rip.

under reduced loads, and then dropped abruptly to near zero upon the first opening transition, as the duplex elements P1 and P2 unfolded. In the presence of TPP (*Figure 3B*), the FRET efficiency of F'•TPP started out with a similar low value, but then transitioned to an intermediate level which was not observed in the absence of ligand, until the elements P1, P2 and P4/5 unfolded together,

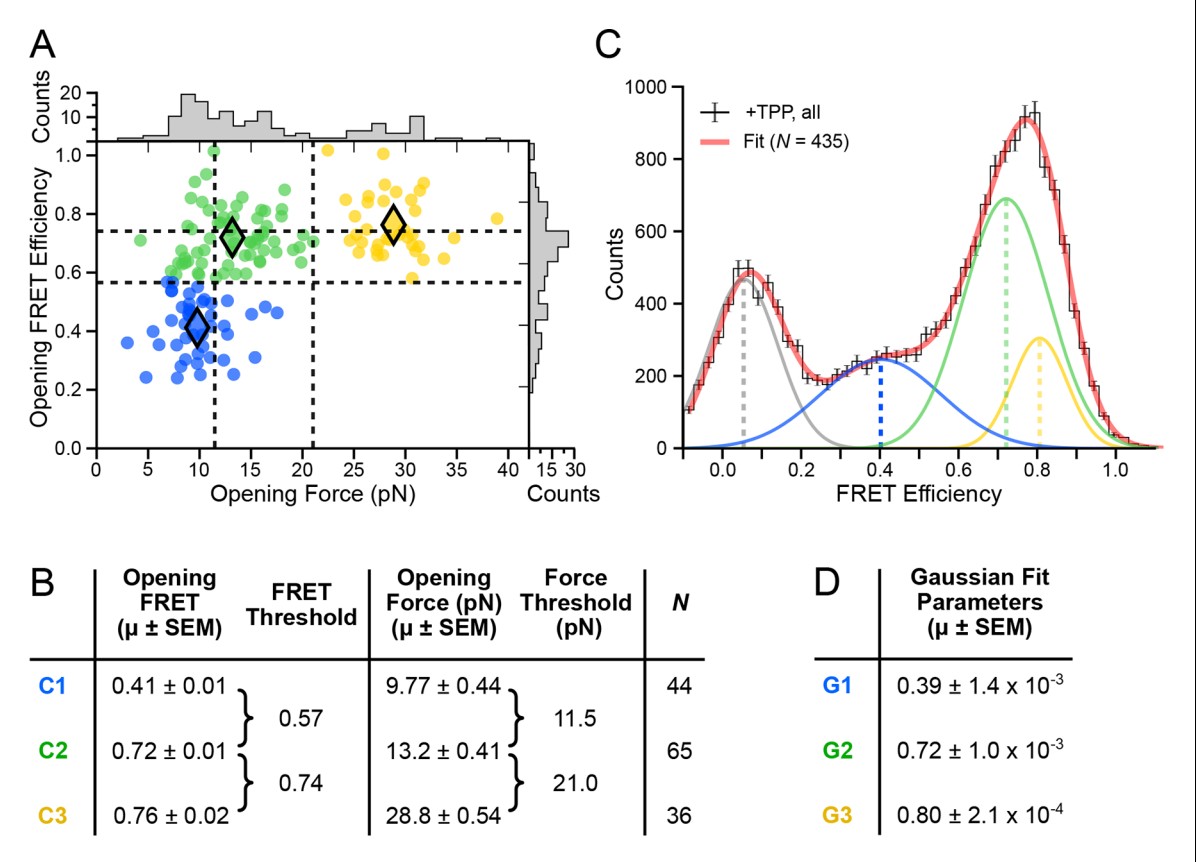

**Figure 4.** Clustering analysis of opening FRET values, and global analysis of full-length FRET traces. (A) *k*-means clustering for opening FRET and opening force values. Filled diamonds mark the mean opening force and opening FRET values for each of the *k* = 3 populations; dashed vertical and horizontal lines indicate the thresholds for force and FRET states, respectively. (B) Table summarizing opening force and FRET centroids, and thresholds for the clusters C1, C2 and C3. (C) Global fit (red) of all data (black histogram, with error bars) to a sum of four Gaussians. Individual Gaussian fits are shown as colored lines, with the mean values indicated (vertical dashed lines). (D) Table of fit parameters for G1, G2 and G3 (N = 435).
The following figure supplement is available for figure 4:

**Figure supplement 1.** *k*-means cluster analysis of opening force and FRET.

resulting in an abrupt drop to near zero. This transition was followed by a very tiny rip, barely detectable in the FEC, but not producing any further change in FRET, that corresponded to the unfolding of P3 (*Anthony et al., 2012*). By contrast, FRET signals for F″•TPP (*Figure 3C*) first jumped from a low to an intermediate FRET level (the same two levels, on average, displayed by F′•TPP), but later transitioned to a high FRET level, characterized by a comparatively low variance, until the final opening transition, corresponding to the unfolding of the entire aptamer, which returned the FRET level to the baseline.

## Cluster analysis establishes thresholds for the three FRET states

In both force and FRET channels, unfolding events produced abrupt drops in signal levels. We recorded the FRET efficiencies prior to such events (defined as the opening FRET values) along with the associated forces (defined as the opening force values). A plot of opening FRET versus opening force revealed sub-populations that were binned by *k*-means cluster analysis (*Figure 4A*); setting *k* = 3 selected distinct clusters and established threshold values for each coordinate axis (*Figure 4B*). Vertical (force) thresholds demark the three conformational forms, based chiefly on their secondary structures: FECs with an opening force lower than 11.5 pN were assigned to aptamers unfolded when not bound to TPP; opening forces between 11.5 pN and 21.0 pN were assigned to aptamers

that unfolded in the F'•TPP state; and opening forces above 21.0 pN were assigned to aptamers that unfolded from the F″•TPP state. The mean opening forces computed for each of the three clusters (C1, C2, and C3) matched, within error, the force values that had been estimated in a previous study using other methods (*Anthony et al., 2012*), validating the choice of k.

Horizontal (FRET) thresholds, 0.57 and 0.74, demark the FRET cutoffs for the clusters. The corresponding FRET sub-populations were then compared to a global histogram of all single-molecule records fitted to a sum of four Gaussians (*Figure 4C*). Mean values returned from the Gaussian fits, G1, G2 and G3, were statistically consistent with mean values for opening FRET computed for the C1, C2 and C3 clusters. By matching the FRET sub-populations against conformations previously characterized by force assignment, we could infer the P3-L5 arm configurations, independent of force. The lowest-valued FRET sub-population, corresponding to the APO state, reflects a significant physical separation between the P3 stem and L5 loop, which remain consistently apart. The intermediate-value FRET sub-population, corresponding to the WB (weakly bound) state, reflects more proximal stem-loop interactions. The highest-value FRET population, corresponding to the SB (strongly bound) state, reflects the close apposition of P3 and L5. To characterize further P3-L5 arm behavior, we used the cluster thresholds established for refolding to categorize segments of FEC records obtained at forces up to 10 pN (i.e., prior to the first opening transition), including the refolding portion, as follows. Segments of FRET trajectories below 0.57, or above 0.74, which tended to exhibit comparatively lower variance, were assigned to the APO or SB states, respectively. FRET trajectories between these values were assigned to the higher-variance, WB state (such trajectories included short intervals where the FRET value made brief excursions above or below the threshold levels, but never remained consistently out-of-range for more than one or two consecutive time points). Thus separated, the individual FRET data were binned into histograms for the corresponding APO, WB, and SB states (*Figure 5A*).

## TPP analogs reveal differences in helix-arm configuration during ligand binding

TPP analog ligands that are missing the terminal phosphate group, such as TMP (thiamine monophosphate) and T (thiamine), are thought to be able to form contacts with only one half of the bipartite binding site of the aptamer. In support of this notion, force spectroscopy has previously established that FECs of the aptamer bound by such analogs closely resemble those corresponding to the weakly bound, F'•TPP conformation (*Anthony et al., 2012*). We collected data for both FECs and FRET trajectories for aptamer binding in the presence of TMP, T, and in the absence of ligand, and applied the same clustering formalism as for aptamer binding in the presence of TPP (*Figure 5C*). FRET trajectories acquired in the absence of ligand exhibited a low efficiency (mean = 0.46), and closely matched those assigned earlier to the APO state, which had been observed in the presence of TPP, but prior to presumed ligand binding, lending additional experimental support to that assignment. Aptamers bound to either TMP or T generated FECs similar to those previously observed, and closely resembled the weakly bound conformation in the presence of TPP, as anticipated. The FRET efficiencies were similar for both T and TMP ligands (means = 0.64 and 0.69, respectively). However, both these values are significantly lower than the efficiency of the WB state obtained in the presence of TPP (mean = 0.73). Because the opening transitions in FECs tend to be dominated by secondary structure, whereas the FRET signal reports a tertiary rearrangement, we conclude that aptamers bound to TMP or T share a similar secondary structure with the aptamer weakly bound to TPP, but differ in tertiary structure: specifically, the analogs TMP and T fail to bring the helix arms together as closely as TPP.

## Hidden Markov modeling of FRET refolding trajectories reveals rates of aptamer helix arm dynamics

To quantify the helix arm dynamics, we fitted a sequential, four-state hidden Markov model (HMM) to refolding portions of the FRET trajectories (obtained under negligible loads that permit structure formation, <5 pN) (*Figure 6A*). FRET values and directional transition rates were obtained at TPP concentrations of both 5 µM and 50 µM (*Figures 6B,C*). The corresponding values for FRET states identified by hidden Markov modeling were identical, within statistical error, at both TPP concentrations, and notably, also within statistical error for the corresponding states identified using k-means

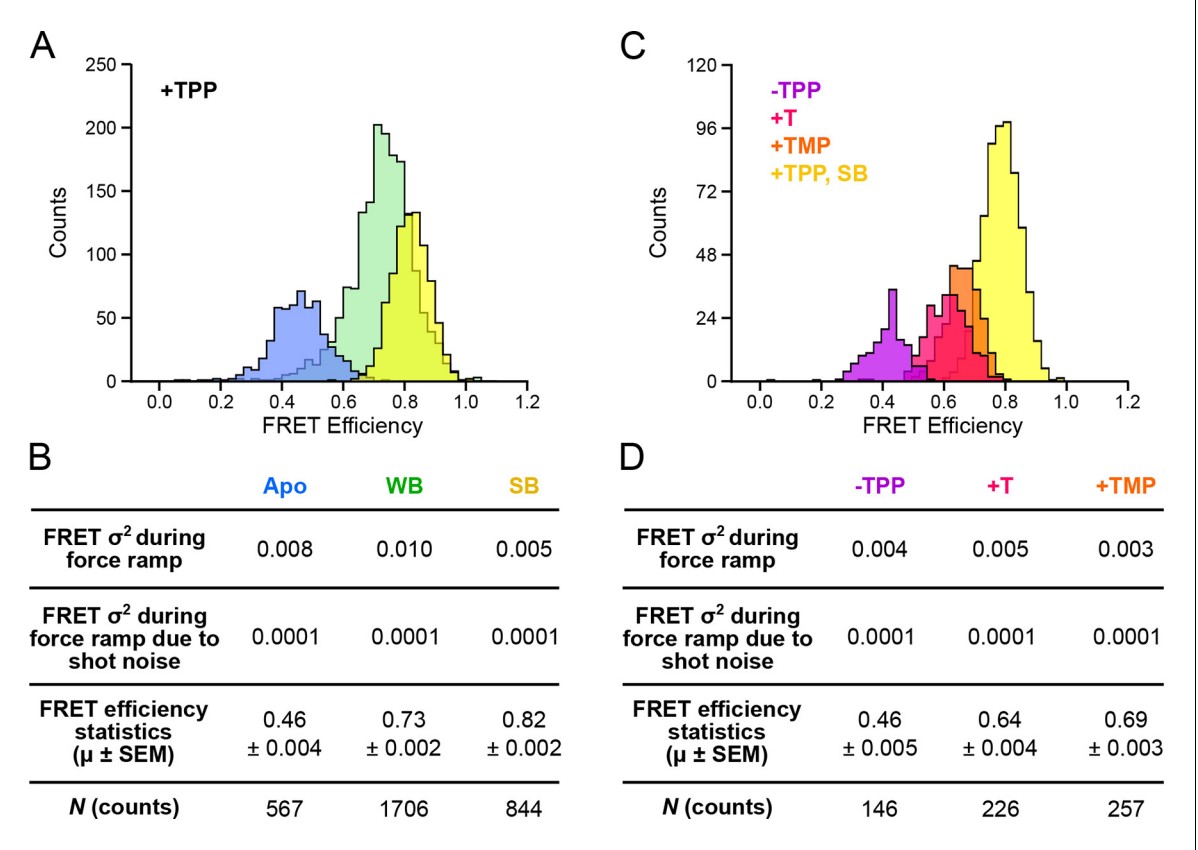

**Figure 5.** Distributions of segmented FRET data in the presence and absence of TPP and its analogs. (**A**) Segmented FRET data in the presence of saturating TPP (2 mM). After segmenting, FRET records were categorized into APO (blue), WB (green), and SB (yellow) FRET states. (**B**) Table summarizing FRET efficiency statistics. (**C**) FRET data from aptamers in the absence of TPP (purple), and in the presence of saturating (2 mM) T (pink) and TMP (orange), displayed together with data for the SB state in the presence of TPP (yellow) for comparison. (**D**) Table summarizing FRET efficiency statistics for the TPP, T, and TMP conditions indicated.

cluster analysis (*Figure 6B* and *Figure 5B*). We were therefore able to make a direct correspondence between the HMM-identified states (called A, B, and C) and the clustered FRET states (Apo, WB, and SB, respectively). At the higher TPP concentration, all transition rates except for the A → B rate remained the same within error, implying that only this transition, corresponding to the Apo → WB transition, is TPP-dependent (*Figure 6C*). The inferred dependence is also consistent with the kinetic model proposed on the basis of force spectroscopy, where only the transition between states F' and F''•TPP is TPP-dependent (*Figure 7A and 7B*) (*Anthony et al., 2012*). Therefore, both FRET and force data support a sequential, four-state kinetic model. However, the transition rates obtained using the HMM approach are consistently faster than the rates previously derived from force spectroscopy alone. Specifically, at the high TPP concentration, the lifetime in the A state was vanishingly short, and only a single dwell interval was scored by the HMM procedure. Instead, the vast majority of records (N = 52 of 53) started out in state B, and thereafter made subsequent transitions between B and C states. We conclude that the A → B transition rate at the higher TPP concentration is likely faster than our detection capability, and that the apparent 4-fold increase in this transition rate should be taken as a lower bound.

## Discussion

The concurrent use of force spectroscopy, which primarily monitors secondary structural states, and FRET, which can probe tertiary structural rearrangements, has helped to reveal new details of the interaction between ligand binding and conformational change in the TPP riboswitch

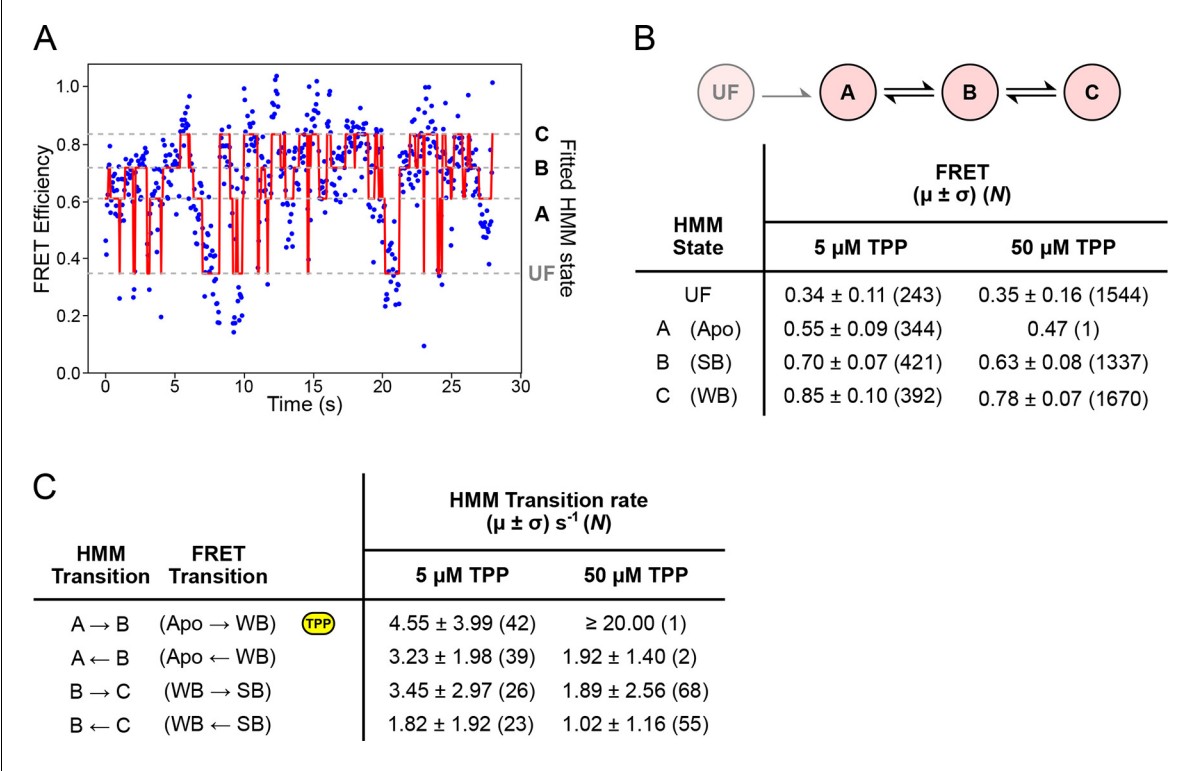

**Figure 6.** HMM modeling of refolding FRET trajectories. (**A**) Four-state HMM fit to a 30-second portion of concatenated refolding FRET trajectories in the presence of 5 µM TPP. (**B**) Top: Reaction diagram for the four-state, sequential HMM model, with states UF and A (donor-fluorescing), and B and C (acceptor-fluorescing). Bottom: table summarizing FRET values obtained 5 µM and 50 µM TPP concentrations for each state. (**C**) Table summarizing directional rates for transitions between HMM states A, B, and C. The yellow icon indicates the TPP-dependent transition rate.

(*Anthony et al., 2012*). The eukaryotic aptamer folds quickly into a stable form, where the P3-L5 arm configuration remains open, as indicated by the stable, low FRET trajectories recorded in the absence of ligand (*Figure 3A*, *Figure 3—figure supplement 2F–H*). By contrast, *Haller et al. (2013)* reported that, in the absence of ligand, the arms of the bacterial form of the TPP aptamer interchange rapidly among three FRET states, as fit by a hidden Markov model, with dwell times around 100 ms (*Haller et al., 2013*). The slightly slower imaging rate employed here (20 frames/s, vs. ~50–100 frames/s should still have been adequate to detect any comparable flickering, in principle, but none was evident in our records. The eukaryotic aptamer is therefore structurally more stable in its unbound form (or conceivably, it switches among configurations, but at a significantly faster rate). We note that the two TPP aptamers differ primarily in their P3 stem regions: the bacterial form lacks a distal P3 extension that is present in the eukaryotic form, and this longer element may exert a stabilizing role on the unbound aptamer.

Upon ligand binding, the P3 and L5 arms underwent a rapid conformational change, as manifested by the FRET signal, corresponding to the transition from the APO to the WB state (*Figures 3B,C*; blue to green boxes). The FRET levels in the WB state exhibited comparatively high variance (*Figure 5B*), and appeared to flicker between intermediate and high FRET values. Transitions out of the WB state were generally irreversible, and these led directly to the SB state, which was characterized by a stable, high FRET level with significantly lower variance. We therefore interpret the SB configuration as one where the two helix arms are fully docked, forming a tight, stable P3-L5 tertiary interaction. Based upon the data, the WB state can be interpreted as one where the P3 and L5 arms are partially docked, that is, they remain mobile but closer, on average, than in the APO state. Alternatively, the WB state might represent a superposition of (less individually flexible) docked and undocked conformations that interchange rapidly on the timescale of our data acquisition, at rates >>20 s$^{-1}$ (*Figure 6C*). Put another way, the WB state must be mobile, and consistent

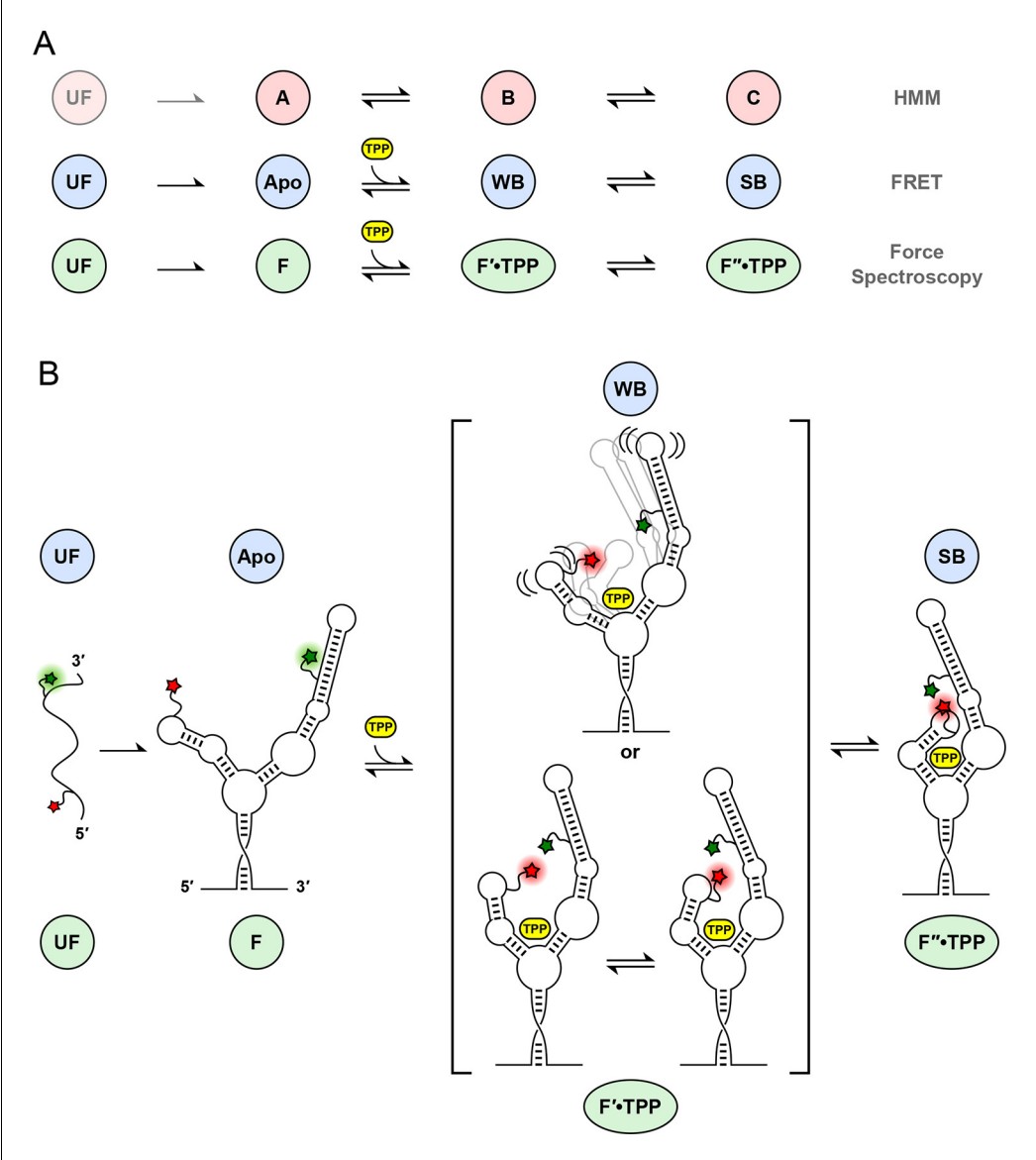

**Figure 7.** Correspondences among single-molecule state identifications, and model of aptamer binding to TPP. (**A**) Cartoon summarizing the correspondence between analysis and data collection methodologies. The corresponding states and reaction schemes obtained from HMM analysis (top line; pink circles), helix-arm configuration states obtained from FRET signals (middle line; blue circles), and secondary structural states inferred from force spectroscopy (lower line; green ellipses) are shown. (**B**) Model for ligand binding and associated conformational changes. Colored labels indicate states based on aptamer secondary structure (green circles and ellipses; derived from force data) and sensor-helix arm configuration (blue circles; derived from FRET data). Prior to TPP binding, the sensor arms are apart. Subsequent to TPP binding, the arms remain mobile but begin to move closer together in the weakly-bound, liganded state. This mobility may reflect a type of conformational heterogeneity that is either dynamic (top), with flexible arms, or static (bottom), with rapid interconversions between transient states (square brackets; see text). The system subsequently transitions, on a timescale of around a second, to a strongly bound state, with the sensor arms fully docked and largely immobilized.

with either dynamic conformational flexibility or with static conformational heterogeneity (*Torella et al., 2011*).

Helix arm behavior was examined further by exploring the effects of aptamer binding by the TPP analogs, thiamine (T) and thiamine monophosphate (TMP). The binding site of the aptamer, located at the central junction of the arms, is bipartite, and is able to bind to the thiamine moiety of TPP on one side, and to phosphate groups on the other. One previous study of T and TMP, which lack one and two phosphate groups, respectively, and therefore bind to only one side of the site, concluded

that the lack of phosphates prevents proper alignment of the P3-L5 arms and ultimately, docking (*Noeske et al., 2006*). For both bound analogs, the mean FRET efficiencies were smaller than the corresponding FRET efficiency of the SB state (*Figure 5D*). These results imply that the sensor helix arms spend most of their time farther apart when bound to TMP or T than when bound to TPP in the F″•TPP conformation. We surmise that each additional phosphate group serves to draw the arms closer together during ligand binding, and is responsible for the observed increase in FRET. Perhaps more surprisingly, the FRET trajectories for the aptamer after binding onto TMP and T are stable and low in variance, suggesting that although the P3-L5 structures are unable to dock, they remain in a stable configuration (*Figures 5C and 5D*). This trend is reflected in the FRET variance obtained during force ramp, which is low at 0.005 and 0.003 for T and TMP, respectively, but significantly larger at 0.010 for the WB FRET state (*Figures 5B and 5D*). This result implies that when the aptamer is bound to T or TMP, the arms are less dynamic than when the aptamer is bound to TPP in the F′•TPP conformation. Taken together, these data suggest that aptamer binding of TPP in the F′•TPP conformation is a state in which the helix arms are in close proximity due to the presence of two phosphate groups, yet exhibit significant flexibility. This flexibility appears to be essential for proper orientation of the helix arms, which primes the arms to dock and the aptamer to adopt the F″•TPP conformation.

Our studies of the complete eukaryotic TPP riboswitch reveal a ligand binding process distinct from that of the bacterial TPP riboswitch. We observed a rather rigid helix arm configuration during the fully folded, APO state, in contrast to the dynamic helix arm behavior reported for the *E. coli* TPP riboswitch in the absence of TPP. Upon TPP binding, we observed an increase in helix arm dynamics in the WB state, which is also in contrast to bacterial studies that observed a reduction of arm dynamics upon ligand binding. Our observations of FRET trajectories suggest that helix-arm fluctuations occur at significantly faster rates than any secondary structural changes. Based on our data, we conclude that both secondary and tertiary interactions are vital to tight TPP binding. We speculate that in the eukaryotic aptamer, ligand binding first anchors the lower regions of the helix arms, inducing strains that lead to fluctuations in the more distal regions. Such fluctuations increase the frequency of contact between the P3-L5 arms, and thereby increase the probability of docking, lowering the binding energy further. Future studies using additional FRET probes, and conducted at even higher time resolution, should be able to probe independently the motions of the lower aptamer regions.

The bacterial and eukaryotic forms of the TPP riboswitch share a similar core structure, bind identical ligands, and regulate thiamine pyrophosphate synthesis (albeit in rather different ways). However, it has become clear that even minor structural differences between forms can yield distinct properties. For example, the comparatively longer and more stable P3 stem found in the eukaryotic riboswitch leads to slower arm dynamics, as expected, with wide-ranging consequences. But piecing together a more detailed, quantitative picture of the folding landscape, which includes strain-dependent ligand binding and the formation of tertiary contacts during helix-arm docking, requires special methodology, such as the combination of fluorescence and force spectroscopy pursued here. It is still early days, but we anticipate that the combined force-FRET technique will be a powerful tool for elucidating riboswitch function across all kingdoms of life.

# Materials and methods

## Sequences used
wtAtRs21 (64 nt) U88U (P3)
AUC GAG AGG GAC ACG GGG AAA CAC CAC CAU AUA U GCA CCA GGG GUG CUU GAA CCA GGA (**5-N-U**)AG CCU

wtAtRs2 (Long Chain 64 nt) U25U (P4/5)
/5Phos/ GCG AAA AGG CGG GCU AUC CGG GAC CAG GCU GAG AAA GUC CCU UUG AAC CUG AAC AGG G(**5-LC-N-U**)A AUG C

wtAtRs3 (53 nt)
/5Phos/ CUG CGC AGG GAG UGU GCU AAC AAG GUC ACC AUC AUC CUG ACU AGA GUC
CUU GG

## DNA splints
P45_splint_60
CTCTAGTCAGGATGATGGTGACCTTGTTAGCACACTCCCTGCGCAGGCATTACCCTGTTC

P3_splint_41
GCCCGCCTTTTCGCAGGCTATCCTGGTTCAAGCACCCCTGG

## Nucleic acid labeling
An instrument characterization/calibration was performed using DNA hairpins with short tetraloops as controls. We examined two hairpins: one with a 15-bp stem (15R50/T4) and one with a 20-bp stem (20R25/T4). The nanomechanical properties of both hairpins have been extensively characterized in a previous force-spectroscopy study (*Woodside et al., 2006*). Hairpins were constructed by ligating separately prepared 5′ and 3′ fragments (*Figure 8A*); in each of the fragments, a single amino-modified thymidine residue was introduced near the base of the hairpin (Integrated DNA Technologies). The modified bases were labeled with the fluorescent dyes, ATTO550 and ATTO647N (ATTO-TEC), using an amine-labeling chemistry (*Kricka, 2002*; *Solomatin and Herschlag, 2009*). The dye-labeled fragments were annealed using a temperature ramp protocol, from 95°C to 4°C in 40 min, and ligated with T4 DNA ligase (Invitrogen, Carlsbad, CA) for 30 min at 37°C (*Akiyama and Stone, 2009*). Proteins were removed by phenol-chloroform extraction, and the ligated hairpins were ethanol-extracted. Doubly labeled hairpins were separated from incomplete hairpin species on a 10% PAGE gel, then electro-eluted, followed by ethanol precipitation.

The TPP riboswitch was split into three fragments: wtAtRs1, wtAtRs2, and wtAtRs3 (Dharmacon, Lafayette, CO). Fluorescent dyes were attached via amino-modified uracil bases at G88 (5-N-U) and U25 (5-LC-N-U). G88 is found in the P3 stem structure of fragment wtAtRs1, and was labeled with ATTO550 dye; U25 is found in the L5 loop of fragment wtAtRs3 and was labeled with ATTO647N dye. Labeled fragments were splint-ligated using T4 RNA ligase 2, assisted by complimentary DNA oligomers that bridged the fragments. Labeled and ligated riboswitches were gel extracted and eluted, as described (*Anthony et al., 2012*), for the DNA hairpins.

## Dye characterization
### Base modifications and dye labeling do not interfere with opening force and opening distance in DNA hairpins
For calibration and confirmation purposes, we conducted a preliminary optical trapping-smFRET study using dye-modified versions of a DNA hairpin which has been extensively characterized, 15R50T4 (here, called SJ1) (*Blakely, et al., 2014*; *Woodside et al., 2006*). Opening force ($F_{1/2}$) and opening distance ($\Delta x$) values obtained during pulling experiments were compared to those obtained at equilibrium under constant force (*Figure 8A*). The $F_{1/2}$ and $\Delta x$ values were also measured for SJ1 hairpins with internal amino-modified thymine base dye labeling sites (Integrated DNA Technologies), and ATTO550-ATTO647N dye-labeled hairpins. Comparisons of $F_{1/2}$ and $\Delta x$ values between unmodified and amino-modified, but unlabeled, SJ1 hairpins confirmed that nucleotide modification and fluorescent labeling have minimal effects on the measured mechanical properties. The minor differences that were observed may be attributable to the fact that comparatively rapid force ramps may slightly overstretch a DNA hairpin just prior to duplex opening, resulting in a small force overshoot, and consequent overestimation of the extension, as noted (*Li et al., 2006*; *Liphardt et al., 2001*).

Previous studies have shown that fluorescent dye labels that get placed very close to neighboring nucleotides can become quenched, especially near guanine (*Di Fiori and Meller, 2010*). To test this observation, we designed DNA hairpin 20R25/4T (SJ2). On SJ2, amino-modified thymine labeling sites were situated three nucleotides from either side of the base of the hairpin stem, and G:C base pairs located near the base, and guanine residues in the adjacent single-stranded regions, were replaced with adenine or thymine. Again, fluorescently-labeled and base-modified SJ2 hairpins

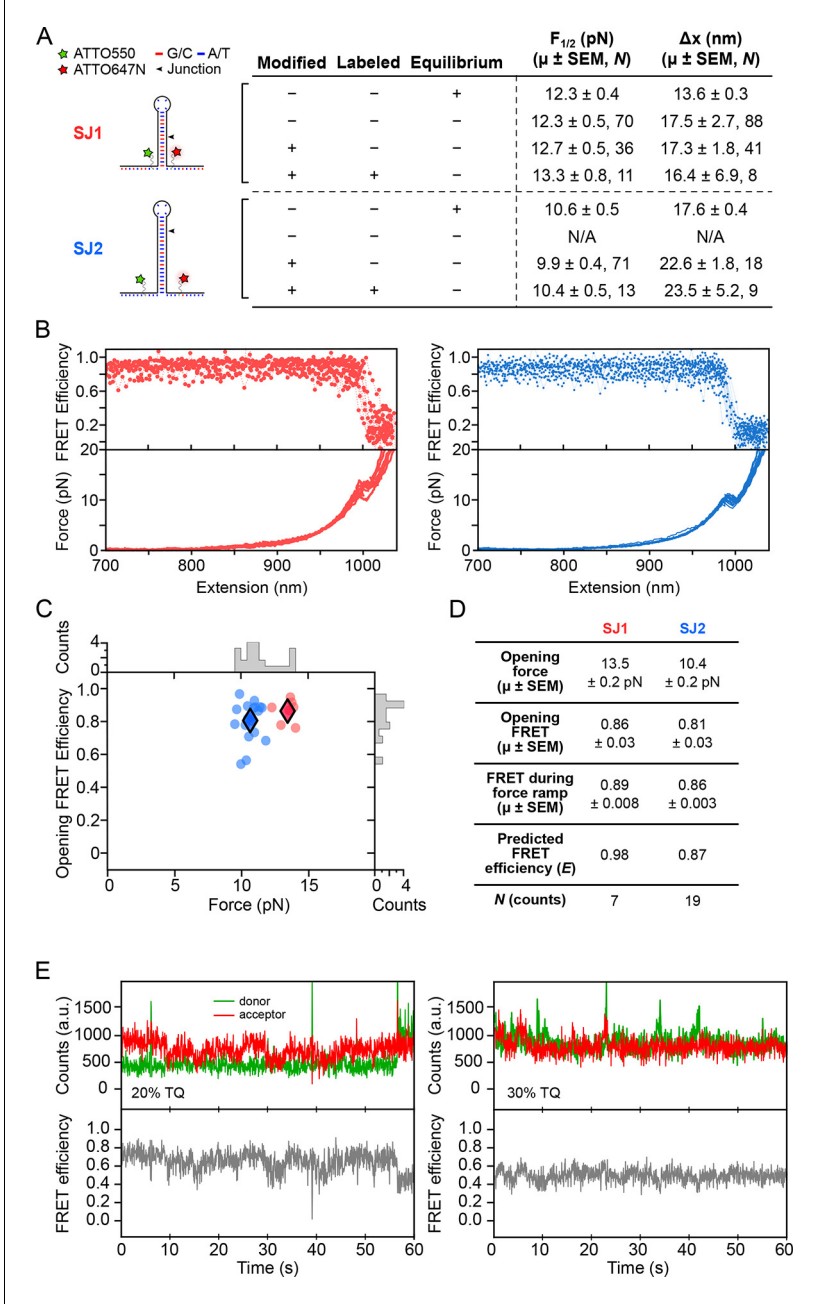

**Figure 8.** Dye characterization using DNA hairpins. (**A**) Opening distances ($F_{1/2}$) and forces ($\Delta x$) of DNA hairpins SJ1 and SJ2 were measured at non-equilibrium and compared to previous measurements at equilibrium. (**B**) FRET trajectories and FECs are shown. (**C**) *k*-means clustering for SJ1 and SJ2 FRET trajectories. Cluster means are indicated with filled diamonds. (**D**) Table summarizing opening force and FRET statistics, force ramp FRET statistics, plus predicted FRET efficiency. (**E**) Donor and acceptor traces (top) in the presence of 20 and 30% TQ (left and right, respectively).

reported $F_{1/2}$ and $\Delta x$ values comparable to those of base-modified and unlabeled SJ2 hairpins, confirming that base modification and fluorescent labeling did not perturb DNA hairpin structure nor folding dynamics (*Figure 8A*).

## FRET measurements agree with predicted FRET values
The FRET readout of the ATTO550-ATTO647N dye pair during non-equilibrium pulling experiments of DNA hairpins was measured with an exposure time of 50 ms. For both SJ1 and SJ2, FRET

transitions from high to low FRET efficiency were temporally coincident with hairpin rupture events (*Figure 8B*). FRET values recorded at the time of hairpin opening matched theoretical FRET efficiency (*E*) calculated from the dipole-dipole coupling mechanism expression $E = 1/[1 + (R/R_0)^6]$, where *R* is the inter-dye distance and $R_0$ is the Förster radius of the ATTO550-ATTO647N dye pair. Opening FRET was plotted against opening force, and *k*-means clustering with *k* = 2 was applied to identify the centroids of each hairpin cluster (*Figure 8C*). The clustering FRET mean for SJ1 (0.89) was significantly lower than the predicted value (0.98) (*Figure 8D*). However, since SJ2 reports a clustering FRET mean (0.86) consistent with predicted FRET (0.87), we believe that the FRET discrepancy for SJ1 can be explained by dye interactions with pyrimidine nucleotides located at the base of the hairpin. The RNA aptamer construct was designed accordingly to avoid such interactions.

## Trolox

Trolox (TX), a soluble analog of vitamin E, has been shown to reduce triplet-state quenching and blinking, especially when combined with Trolox-quinone (TQ) (*Cordes et al., 2009*; *2011*). TX was dissolved in DMSO to 100 mM and exposed to UV light, yielding 20–28% TQ in 3 mM TX. The TQ concentration was determined by diluting UV-exposed TX to 3 mM in PHC buffer (50 mM HEPES pH 7.5, 130 mM KCl, 4 mM $Mg^{2+}$, 1 mM EDTA) and measuring the absorbance at 255 nm, according to:

$$[TQ] = \frac{A_\gamma/d - \varepsilon_{\gamma TX}[TX]_0}{\varepsilon_{\gamma TQ} - \varepsilon_{\gamma TX}}$$

where γ is the wavelength (255 nm), *d* is the distance (1 cm), and the extinction coefficients for TX ($\varepsilon_{\gamma TX}$) and TQ ($\varepsilon_{\gamma TQ}$) are 400 and 11,600 $l \cdot mol^{-1} \cdot cm^{-1}$, respectively. Lower percentages of TQ were obtained by diluting TQ in 100 mM unexposed TX. A stock was stored at 4°C, and was diluted to 3 mM in SCAV prior to experiments. DNA hairpins held under low load showed reduced blinking in the presence of TQ (*Figure 8E*).

## Simultaneous single-molecule Förster resonance energy transfer optical trapping instrumentation and data collection

We incorporated fluorescence excitation and detection capabilities into an existing dual-beam optical trapping apparatus, as described (*Greenleaf et al., 2008*; *Lang et al., 2002*). Briefly, the setup (schematic; *Figure 2*) is based on a Nikon TE2000 inverted light microscope (Nikon Instruments, Melville, NY), and utilizes three lasers: one for optical trapping (10W; 1,064 nm; diode-pumped Nd: YVO$_4$ laser; Spectra-Physics Lasers, Mountain View, CA), one for position detection (20 mW; 830 nm; single-mode diode laser; Qioptiq, Waltham, MA), and one for fluorescence excitation (1 W; 532 nm; diode-pumped, frequency-doubled YAG; CrystaLaser, Reno, NV). Fluorescence emission was detected by an electron-multiplying CCD (EMCCD) camera used for imaging (iXon 897; Andor, United Kingdom). Fluorescence excitation light was coupled into, and focused through, the microscope objective to a ~1 μm spot in the specimen plane, confocal with the fluorescently-labeled RNA construct. Dichroic filters were installed upstream of the EMCCD to split the emission spectra of the donor and acceptor fluorophores into spatially separate channels at the image plane; the emission image was passed through a rectangular slit to allow simultaneous collection from both acceptor and the donor fluorophores.

The fluorescence sub-image size was ~43 x 283 pixels (pixel size, 0.16 x 0.16 μm). Each region of interest (ROI) subtended 7 x 10 pixels for both the donor and acceptor channels. The EMCCD output counts for ATTO550 and -647N dyes were typically around 16,000 and 7,000 photons per second, respectively. Successive frame exposure times were 50 ms, and about 800 frames were collected per single-molecule record.

## Dumbbell assay

Polystyrene beads, 0.6 and 0.8 μm in diameter, were functionalized with streptavidin and anti-digoxigenin and coupled to 1 kb and 2 kb DNA handles, respectively, via single-stranded overhangs, as described (*Anthony et al., 2012*; *Greenleaf et al., 2008*). Handle lengths were chosen to provide adequate spatial separation (approximately 1 μm) between the optical trapping beams and the fluorophore-labeled constructs. Constructs for study (DNA hairpins or RNA aptamers) were first

annealed to the handles, and then conjugated to beads by incubation for 1 hr at room temperature. The bead incubation was diluted in oxygen-scavenging buffer (SCAV; 50 mM HEPES pH 7.5, 130 mM KCl, 4 mM $Mg^{2+}$, 1 mM EDTA, 160 Units/ml glucose oxidase, 100 Units/ml catalase, 0.8% beta-glucose, and 3 mM UV-treated Trolox (Sigma-Aldrich, Germany) with 10–14% Trolox-quinone). Both glucose oxidase and catalase were further purified from manufacturer's stocks by ultrafiltration. Samples were then introduced into flow cells for optical trapping experiments.

### Data acquisition and analysis

FECs were obtained by servoing the steerable optical trap at a constant rate ($\sim$90 nm s$^{-1}$; stiffness 0.25–0.3 pN nm$^{-1}$) relative to the fixed trap, using an acousto-optic deflector. Opening forces and extensions were measured by first identifying transitions in the FECs, then fitting the regions before and after these to a double wormlike chain model (WLC), as described (*Greenleaf et al., 2008*). For smFRET traces, correction factors for the leakage of donor emission into the acceptor channel ($\beta$) and differences between the donor and acceptor in efficiency and quantum yield ($\gamma$) were calculated as described (*McCann et al., 2010*). The $\beta$ value was calculated to be 0.13 for the ATTO550-ATTO647N dye pair. The $\gamma$ corrections varied among traces, and were determined by subtracting counts before and after the FRET transition for both donor and acceptor signals. After suitable $\beta$, $\gamma$, and background corrections, the FRET efficiencies were calculated from $E = I_A /(I_A + I_D)$.

$k$-means clustering for opening forces and FRET values was performed using Python libraries. The clustering parameter, $k$, was varied from 2–4 with default settings (*Figure 4—figure supplement 1*), with optimal results obtained for $k = 3$. FRET data were binned using a bin size of 0.025. The global histogram of FRET data was fit to a sum of four Gaussians using a conventional nonlinear least-squares algorithm.

Refolding FRET traces were concatenated and analyzed using the MSMBuilder HMM Python package (*Beauchamp et al., 2011*). FRET values and inter-state transition times were calculated for HMM states A, B and C using averaging. Transition rates were calculated from the reciprocals of the directional transition times.

## Acknowledgements

We thank Dr. Anirban Chakraborty for helpful comments.

## Additional information

### Funding

| Funder | Grant reference number | Author |
|---|---|---|
| National Institute of General Medical Sciences | GM057035 | Steven M Block |

The funders had no role in study design, data collection and interpretation, or the decision to submit the work for publication.

### Author contributions

VKD, Conception and design, Acquisition of data, Analysis and interpretation of data, Drafting or revising the article; ITFH, Acquisition of data, Analysis and interpretation of data, Drafting or revising the article; CFP, Conception and design, Acquisition of data, Analysis and interpretation of data; DWH, Analysis and interpretation of data, Drafting or revising the article; SMB, Conception and design, Analysis and interpretation of data, Drafting or revising the article

### Author ORCIDs

Van K Duesterberg, http://orcid.org/0000-0002-6036-5453

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
