## [Decision Letter]

Thank you for submitting your work entitled "Observation of long-range tertiary interactions during ligand binding by the TPP riboswitch aptamer" for consideration by *eLife*. Your article has been reviewed by three reviewers, one of whom is a member of our Board of Reviewing Editors, and the evaluation has been overseen John Kuriyan as the Senior Editor.

The reviewers have discussed the reviews with one another and the Reviewing Editor has drafted this decision to help you prepare a revised submission.

Summary:

By combining optical tweezers with FRET, the authors visualize the different conformational transitions that take place when the *A. thaliana* thiamine pyrophosphate (TPP) riboswitch unfolds. Single-molecule measurements can reveal hidden details – especially transient dynamics and rarely populated states – and previous single-molecule studies of TPP and other riboswitch aptamers using either optical trap or single-molecule FRET have already provided novel insights. Optical traps allow the application of mechanical forces to the ends of the aptamer and measure the forces at which unfolding and folding occur. Based on the extension change during such 'rips', the local structures that are being altered can be characterized and assigned. But because the optical trap alone reports only on the end-to-end distance of the aptamer, the internal degrees of freedom cannot be examined, at least not directly. In smFRET, different parts of the molecule can be labeled so that docking and undocking of part of the molecule that is important for the ligand-specific structural organization and is distant from the primary stem can be studied. By adding the smFRET capability to the optical trap setup, they are able to ask whether FRET changes can occur coincident or independent of major rips detected by optical trap.

Using optical tweezers to pull apart the riboswitch, the authors demonstrate the presence of a number of distinct conformations (in line with their previous 2012 PNAS paper). A simultaneous observation of distinct FRET levels indicates that three conformers exist, corresponding to the apo state, a weakly bound form, and a tightly bound one. The use of TPP analogs lacking the terminal phosphate suggests that these ligands only support the weak-binding configuration and that the terminal phosphate is required to bring the two helix arms of the switch together.

Overall, the manuscript describes a well-executed and thoughtful analysis of the function of a eukaryotic TPP riboswitch. This work has been conducted by a top laboratory; the results and conclusions are both sound and compelling. The strength of this manuscript is that it is a wonderfully written description of the function of a TPP riboswitch that likely constitutes the most detailed understanding of these RNAs that has been published to date. The methods used by the authors have revealed information on how ligand binding affects key tertiary interactions, which would be difficult to do by using any other methods.

Essential revisions:

All three reviewers express excitement about the technical tour de force that enabled the work and about the level of detail that is obtained from the simultaneous observation of the dynamics of secondary and tertiary structure formation. However, they raise substantial issues that should be addressed before a revised manuscript can be considered. In particular, a revised manuscript will need to include FRET experiments at zero force to address points number 1 and 2 in the list below. While no additional experiments are required to address the remaining points 3-10, a number of them relate to the need for additional clarification, analyses, and experimental detail. These issues should all be addressed in a revised manuscript and detailed in your rebuttal.

1) The authors claim that their eukaryotic riboswitch behaves differently from the bacterial TPP riboswitch studied by the Blanchard laboratory using smFRET. They, however, do not provide data equivalent to the Blanchard study, that is, no smFRET data at zero force. TIRF-based smFRET analysis is easy to perform and in fact there are several labs at Stanford who can run such an experiment as a control.

2) The authors should present donor and acceptor time traces for at least a couple of traces. The correction factor attributed to the difference in quantum yield is 0.17, suggesting about a factor of six difference. That's surprisingly large. The reference for the correction factor must be incorrect because I could not find the definitions of β and γ in McKinney et al. cited. It is also known that ATTO647N undergoes spectral fluctuations, giving rise to intensity fluctuations on its own, and this can give rise to apparent FRET changes. Extended FRET trajectories at zero force can reassure the readers if this is not a problem.

3) The mechanistic details of one riboswitch example might not be relevant to the mechanistic details of another. Therefore, it is not certain that the knowledge gained from such a detailed study of this TPP riboswitch construct will serve as universal truths and teach us about the function of other members of the same class. Specifically, the authors contrast their findings on a eukaryotic RNA with the previous findings from a bacterial RNA. Unfortunately, their data do not reveal whether these differences are inherent to these divisions of life, or if these differences are just individual characteristics that might vary between any two riboswitches. The authors are well aware of this concern based on some of their comments in the Discussion section. They, however, should discuss the issue of generalizable differences and/or similarities between the classes of riboswitches more explicitly.

4) Hidden-Markov modeling was used to fit their data at forces below 10 pN but the force will certainly influence the observed kinetics. The authors need to show smFRET traces at defined forces and HMM fit overlaid over the raw data. In the current manuscript, they do not show any smFRET data used for HMM analysis and as a consequence, the readers cannot get a feel for how noisy the data is and how reasonable those fits are. In fact, from the current manuscript, I cannot tell whether the smFRET traces used for HMM analysis were obtained at a constant force or during the force ramp.

5) The authors write that the rates determined using HMM analysis of smFRET data are higher than what they saw using optical trap data previously. There are a couple of peculiarities in their presentation. First, in the Discussion section, they say that one is faster than the other but the difference is within error. Well, if the difference is within error, that means the difference is not big enough to be considered significant. Then, why mention it at all? Second, because they acquired optical trap data simultaneously, they should analyze the mechanical data for the transition kinetics and make a direct comparison to rule out the possibility the aptamer modification and fluorescent labeling affect the kinetics even though they do not seem to affect the forces at which the major rips occur.

6) The section “TPP analogs reveal differences in helix-arm configuration during ligand binding”page describing the different FRET levels for different ligands is puzzling. The authors claim that the difference in FRET levels between the WB state for TPP (0.73) and that for TMP or T (0.64 and 0.69, respectively) is significant. However, earlier in the same section, the authors state that a FRET level of 0.46 for traces in the absence of ligand 'closely matches' the 0.41 level observed for the apo state. Why is the one difference relevant and the other not? From the reported SEM's and σ^2 values it is not clear to me why these differences are interpreted differently.

7) The section describing the unfolding features in the FEC's (Figure 2) is very unclear. In the section “Direct correspondence between P3-L5 configuration and secondary structural conformation”, the authors describe the observation of three opening transitions in the absence of TPP and one or two additional features in the presence of ligand. From the data represented in Figure 2 and Figure 2—figure supplement 1, it is not clear to me where these transitions are and how frequently they occur. Reading back the 2012 Anthony et al. paper, I see where they are supposed to occur (for example, Figure 1 in the PNAS paper), but in the current manuscript these observations are poorly detailed. Also confusing is that the assignment of the 'rips' is different in the current manuscript than those in the PNAS paper: the authors attribute the transitions to the structural elements P1, (P2+P4/P5), and P3, while in the PNAS manuscript, these transitions are interpreted as corresponding to the elements (P1+P2), P4/P5, and P3.

8) I do not understand why a single step or two-step FRET increase occurs as the force is ramped up. Are these increases caused by force increase? It must be the case because they do not see such FRET increases during the refolding phase when the force was kept low (~2.5 pN). But I do not understand why higher forces applied would induce L5 and P3 to approach each other. If this can be shown to be the case, I think that is quite surprising and significant. Or perhaps the ligand simply does not bind at 2.5 pN? What happens if they keep the force near zero for an extended period? Would they not see such FRET increases?

9) The authors report a very short lifetime of state A in the presence of high concentration of TPP (50 µM) and state that only one very short-lived event was observed. However, panel F in Figure 2—figure supplement 1 clearly shows the molecule in the low FRET state for almost a second after refolding. The authors should clarify.

10) Figure 4 reports on the σ^2 values of the different FRET states, suggesting that the σ between the apo and SB state only differs by 20-30%. However, visual inspection of the traces in Figure 2 suggests a much more marked difference. Are the values in Figure 4 obtained from a distribution containing the distributions of FRET values for all molecules, or are the σ's obtained at level of individual traces?

---

## [Author Response]

Essential revisions:

All three reviewers express excitement about the technical tour de force that enabled the work and about the level of detail that is obtained from the simultaneous observation of the dynamics of secondary and tertiary structure formation. However, they raise substantial issues that should be addressed before a revised manuscript can be considered. In particular, a revised manuscript will need to include FRET experiments at zero force to address points number 1 and 2 in the list below. While no additional experiments are required to address the remaining points 3-10, a number of them relate to the need for additional clarification, analyses, and experimental detail. These issues should all be addressed in a revised manuscript and detailed in your rebuttal.1) The authors claim that their eukaryotic riboswitch behaves differently from the bacterial TPP riboswitch studied by the Blanchard laboratory using smFRET. They, however, do not provide data equivalent to the Blanchard study, that is, no smFRET data at zero force. TIRF-based smFRET analysis is easy to perform and in fact there are several labs at Stanford who can run such an experiment as a control.

We acknowledge this concern, and we wish to reassure the reviewers at the outset that we have responded proactively by providing new experimental data in our revised manuscript that directly address the issue raised here. But our elaborated response comes in several parts, as follows:

A) We wish to stress that none of the central conclusions of our paper in any way depends upon a comparison made between the findings of the Blanchard lab (relating to a prokaryotic form of the TPP riboswitch) and our own (relating to a eukaryotic form). This is something of an apples‐and‐oranges comparison, in truth: the RNA sequences of these riboswitches differ in significant ways, and we never anticipated that our results would be similar. And indeed, *they were not*. We cited the previous Haller, et al. (2013) findings primarily to point out how different that behavior was from what we observed. Our study was never intended to replicate aspects of the Blanchard group experiments. We employed an entirely different methodology, we investigated a different biological system, and we addressed some very different questions using force spectroscopy. We do not believe that we should be required to replicate the Blanchard experiments using their chosen methodology. Doing so is not feasible (see the reason below), and is very clearly outside the scope of the present study. Regardless of what one might, or might not, learn by so doing, it would not change any of the findings reported in our manuscript, which relate to the eukaryotic form of the riboswitch. The scholarly thing to do is to compare our findings with this other system in the Discussion, and to cite the relevant work, just as we do. We should not be asked to replicate their experiments using their methods. This is akin to asking a crystallographer to acquire an NMR structure, as well.

B) That said, it’s quite clear from the context here that the reviewers are specifically interested in knowing what happens to the eukaryotic TPP riboswitch under “zero force.” That is a perfectly legitimate question. Fortunately, we can answer it without replicating the Blanchard experiments, and we believe that this should satisfy the concern raised. (N.B: The reviewers’ term “zero force” is something of a misnomer, however, because RNA that folds in buffer under zero externalload is by no means subjected to “zero force:” internally, there are some very significant thermal forces, viscous forces, electrostatic forces, and entropic forces to contend with.) The Haller et al. (2013) work was performed without applying external loads. The aptamer dynamics that we observe in our apparatus under the very lowest applied loads, i.e., performed under negligible external forces, are essentially equivalent: these data are not anticipated to differ in any significant way from the results of a TIRF‐based, smFRET study. Specifically, whenever we allow our RNA constructs to refold, we place the molecules under negligible loads (in practice, such forces are all below 5 pN).

Two previous studies on riboswitches (Anthony et al., 2012; Greenleaf et al., 2008) have shown unequivocally that when an RNA construct is held under weak forces, below those required to denature any secondary structure, its folding kinetics are unaffected, and that a riboswitch will quickly regain its fully folded, functional state. These established results imply that any dynamics observed during the refolding process (under <5 pN) should be comparable to the dynamics one would observe at further‐reduced force, down to 0 pN. Put another way, there is no evidence that riboswitch folding changes under the lowest applied loads, between 0‐5 p pN, because RNA secondary structure forms freely within this range. (Incidentally, this makes perfect sense from an energetic perspective, because a 2Å displacement under 5 pN only supplies 1 pN∙nm, which is less than ¼kT!)

C) In light of the foregoing, we’ve collected and analyzed several additional examples of data from refolding trajectories, to address the question about what happens to the riboswitch under negligible external loads. We present these new data, for direct comparison with the Blanchard work, in an additional figure in our revised manuscript. Figure 3—figure supplement 1 shows representative FRET trajectories collected in the absence of the TPP substrate. During the refolding period (0.5 s or 1.0 s), the denatured aptamer is maintained at a negligible load and allowed to refold. The data show that in the absence of TPP, the aptamer adopts the fully‐folded, non‐binding (apo) configuration. This same conformation is stable throughout the subsequent force ramp, right up until the point of the first rip. With the eukaryotic aptamer is in this state, rapid “flickering” transitions into and out of additional FRET states, similar to those reported by Haller, et al., for the bacterial form of the aptamer, *were not observed*. (As we discussed in the original manuscript, our imaging time resolution is slightly lower than that of the Blanchard group, but it’s more than sufficient to have seen comparable fluctuations among states, had these been present.) We draw this comparison not to refute the results of the Haller, et al. (2013) study, but merely to highlight the divergent behaviors between the eukaryotic and bacterial TPP riboswitches.

D) Finally, we strongly dissent from the reviewers’ opinion expressed that “TIRF‐based smFRET analysis is easy to perform.”On the contrary, TIRF‐based smFRET is technically demanding work, and it requires rather specialized apparatus, as well as specialized constructs. Beyond that, it is non‐trivial to conduct the necessary instrument calibrations and data analysis. As a practical matter, our Force‐FRET apparatus, which employs confocal illumination, cannot be used to conduct TIRF‐based smFRET without re‐building its illumination optics altogether, which would render it useless for much of its present work! A different suite of support programs for acquiring and processing images would also have to be written. Even if we were to seek collaboration with some other lab to use their smTIRF apparatus (which we should not be required to do!), the RNA constructs that we developed for use in our optical‐ trapping dumbbell assays are unsuitable for tethering directly to surfaces in smTIRF assays. An entirely different set of molecular constructs would have to synthesized, with different handle lengths and different chemical linkers, suitable for surface‐tethering. Finally, these new constructs would all have to chemically labeled with fluorophores in the lab, assayed for chemical yield, and subsequently tested. Appropriate experimental controls (both positive and negative) would have to be run. This enterprise represents an entire one‐ to two‐year project! Fortunately, it is also completely unnecessary, because we are able to answer the question posed about what happens at negligible forces using data we’ve already acquired in our Force‐FRET study, rendering this possibility moot.

2) The authors should present donor and acceptor time traces for at least a couple of traces. The correction factor attributed to the difference in quantum yield is 0.17, suggesting about a factor of six difference. That's surprisingly large. The reference for the correction factor must be incorrect because I could not find the definitions of β and γ in McKinney et al. cited. It is also known that ATTO647N undergoes spectral fluctuations, giving rise to intensity fluctuations on its own, and this can give rise to apparent FRET changes. Extended FRET trajectories at zero force can reassure the readers if this is not a problem.

We apologize for the confusion here, apparently generated by a misstatement on our part, now corrected: in our manuscript, β refers to the correction that accounts for the leakage of donor emission into the acceptor channel, and not to a “correction for the difference in quantum yield.” We measured a β factor of 0.13. The reference to McKinney et al. was incorrect; the proper reference is to (McCann et al., 2010). The revised manuscript has now been corrected, and we thank the reviewers for calling our attention to this error.

As for ATTO647N, it has been shown to undergo spectral fluctuations when attached to dsDNA, as reported by (Di Fiori and Meller, 2010), whom we cite. Several previous studies have shown that the presence of >1 mM Trolox abolishes the intensity fluctuations from ATTO647N (blinking events) more‐or‐less completely: see, for example, (Cordes et al., 2009). Using DNA hairpin constructs, we collected both donor (ATTO550) and acceptor (ATTO647N) trajectories at constant, low forces in the presence of 3 mM Trolox containing 20% or 30% trolox‐quinone (TQ), respectively (Figure 8). These trajectories show that the increased TQ levels result in FRET records that are virtually free of dye‐blinking events. The absence of any such blinking is also apparent in the negligible‐force refolding data, presented in new Figure 3—figure supplement 1 of the revised manuscript.

3) The mechanistic details of one riboswitch example might not be relevant to the mechanistic details of another. Therefore, it is not certain that the knowledge gained from such a detailed study of this TPP riboswitch construct will serve as universal truths and teach us about the function of other members of the same class. Specifically, the authors contrast their findings on a eukaryotic RNA with the previous findings from a bacterial RNA. Unfortunately, their data do not reveal whether these differences are inherent to these divisions of life, or if these differences are just individual characteristics that might vary between any two riboswitches. The authors are well aware of this concern based on some of their comments in the Discussion section. They, however, should discuss the issue of generalizable differences and/or similarities between the classes of riboswitches more explicitly.

The reviewers make a perfectly valid point, and we entirely agree that the mechanistic details of one riboswitch example might not be relevant to the mechanistic details of another. As a matter of scientific principle (not to mention logic), generalizations shouldn’t ever be attempted until a sufficient number of individual cases has been scored. Clearly, it’s early days yet for these riboswitches, and we know the folding‐energy landscapes for only a mere handful of them. But at the reviewers’ request, we’ve now added our (admittedly, speculative) thoughts on the matter to the end of the Discussion section in the revised manuscript.

*4) Hidden-Markov modeling was used to fit their data at forces below 10 pN but the force will certainly influence the observed kinetics. The authors need to show smFRET traces at defined forces and HMM fit overlaid over the raw data. In the current manuscript, they do* not show *any smFRET data used for HMM analysis and as a consequence, the readers cannot get a feel for how noisy the data is and how reasonable those fits are. In fact, from the current manuscript, I cannot tell whether the smFRET traces used for HMM analysis were obtained at a constant force or during the force ramp.*

We take your point. In the revised manuscript, we now supply the requested overlay of FRET records and HMM fits (please see Revised Figure 6). The FRET‐state assignments, described in the Results sections “Cluster Analysis establishes threshold for the three FRET states” and “TPP analogs reveal differences in helix‐arm configuration during ligand binding,” were performed on data obtained at forces below 10 pN, as stated in the original manuscript. However, the hidden Markov (HM) analysis had been incorrectly described in the Results section “Hidden Markov Modeling of FRET refolding trajectories reveals rates of aptamer helix arm dynamics.” HM modeling was carried out on the refolding portions only of the FRET trajectories. This description has been corrected in the revised manuscript.

5) The authors write that the rates determined using HMM analysis of smFRET data are higher than what they saw using optical trap data previously. There are a couple of peculiarities in their presentation. First, in the Discussion section, they say that one is faster than the other but the difference is within error. Well, if the difference is within error, that means the difference is not big enough to be considered significant. Then, why mention it at all? Second, because they acquired optical trap data simultaneously, they should analyze the mechanical data for the transition kinetics and make a direct comparison to rule out the possibility the aptamer modification and fluorescent labeling affect the kinetics even though they do not seem to affect the forces at which the major rips occur.

The reviewers are absolutely correct to raise this point, and we apologize for any confusion that the offending sentence may have sown. Yes, the difference that we’d noted was within experimental error, and was therefore not statistically significant. As this sentence contributed little to our discussion, it has been removed altogether from the revised manuscript.

*6) The section “TPP analogs reveal differences in helix-arm configuration during ligand binding” describing the different FRET levels for different ligands is puzzling. The authors claim that the difference in FRET levels between the WB state for TPP (0.73) and that for TMP or T (0.64 and 0.69, respectively) is significant. However, earlier in the same section, the authors state that a FRET level of 0.46 for traces in the absence of ligand 'closely matches' the 0.41 level observed for the apo state. Why is the one difference relevant and the other not? From the reported SEM's and* σ*^2 values it is not clear to me why these differences are interpreted differently.*

Yes, the FRET level of 0.46 (± 0.005 s.e.m.) for records obtained in the absence of ligand (the -TPP state FRET efficiency; see revised Figure 4) is said in the section “TPP analogs reveal differences in helix-arm configuration during ligand binding” to “closely match” the FRET level observed in the Apo state (the Apo state FRET efficiency; see revised Figure 4), which is *also* 0.46 ( ± 0.005 s.e.m.) – and *not* 0.41, as the reviewers stated above. Indeed, these two mean values are identical, although they have different std. errs. On the other hand, the WB state for TPP (FRET = 0.73) is a larger value than the WB state for either TMP or T (FRET = 0.64 and 0.69), and the FRET difference (Δ= 0.04‐0.09) is nearly tenfold the std. err. associated with either state (0.004 or 0.003). So, even allowing for systematic errors, this difference is significant.

7) The section describing the unfolding features in the FEC's (Figure 2) is very unclear. In the section “Direct correspondence between P3-L5 configuration and secondary structural conformation”, the authors describe the observation of three opening transitions in the absence of TPP and one or two additional features in the presence of ligand. From the data represented in Figure 2 and Figure 2—figure supplement 1, it is not clear to me where these transitions are and how frequently they occur. Reading back the 2012 Anthony et al. paper, I see where they are supposed to occur (for example, Figure 1 in the PNAS paper), but in the current manuscript these observations are poorly detailed. Also confusing is that the assignment of the 'rips' is different in the current manuscript than those in the PNAS paper: the authors attribute the transitions to the structural elements P1, (P2+P4/P5), and P3, while in the PNAS manuscript, these transitions are interpreted as corresponding to the elements (P1+P2), P4/P5, and P3.

To better highlight the locations of the tiny, individual opening transitions (three transitions in the absence of TPP, and either one or two transitions in the presence of ligand) in the different FECs, we’ve now introduced a set of filled arrowheads into revised Figure 3 (formerly, Figure 2). In the Results section “Direct correspondence between P3-L5 configuration and secondary structural conformation”, the word “additional” refers to the FEC forms. In the absence of TPP, we observed a single form of the FEC. In the presence of TPP, we observed two additional forms. In total, therefore, there are three FEC forms observed.

We thank the reviewers for pointing out the inconsistency regarding the attribution of FEC transitions to denaturing of secondary structures: this has been fixed in the revised manuscript, in the aforementioned section.

8) I do not understand why a single step or two-step FRET increase occurs as the force is ramped up. Are these increases caused by force increase? It must be the case because they do not see such FRET increases during the refolding phase when the force was kept low (~2.5 pN). But I do not understand why higher forces applied would induce L5 and P3 to approach each other. If this can be shown to be the case, I think that is quite surprising and significant. Or perhaps the ligand simply does not bind at 2.5 pN? What happens if they keep the force near zero for an extended period? Would they not see such FRET increases?

Please allow us to explain. If, during a force ramp, the currently applied force happens to be below the value of the opening (“rip”) force for unfolding P1 (typically, <8 pN), then the aptamer behaves just as though it’s under no load (i.e., under negligible force). That is, it’s free to fold or adopt different secondary structures, and to form tertiary contacts. Therefore, any FRET activity monitored within this same force regime may be directly attributed to conformational rearrangement within the aptamer, and it is not caused by the application of force.

FRET increases only tend to occur during the refolding portions of the experiment (Figure 3—figure supplement 2). In most cases (for example, in Figure 3), we observe FRET increases concurrent with force ramps because we allotted only about one second for refolding, whereas arm movements can take place on a much longer timescale. For a counter‐example, Figure 3—figure supplement 1 illustrates an example where riboswitch refolding and ligand binding occurred almost instantaneously, and so we see a variety of FRET transitions during the long refolding time of 3 seconds, and before the force ramp even begins.

9) The authors report a very short lifetime of state A in the presence of high concentration of TPP (50 µM) and state that only one very short-lived event was observed. However, panel F in Figure 2—figure supplement 1 clearly shows the molecule in the low FRET state for almost a second after refolding. The authors should clarify.

The observation of "only one very short‐lived event" came from the results of HMM analysis. In the revised manuscript, Figure 3—figure supplement 1 panel E (previously Figure 2—figure supplement 1 panel F) and panels F‐H supply examples of trajectories that appear, on casual inspection, to be in a low‐FRET state for a period of time, but this interval was actually scored by HMM analysis to be in the mid‐FRET state. In this case, we’re less inclined to trust our eyes and more inclined to trust a less biased analysis.

*10) Figure 4 reports on the* σ*^2 values of the different FRET states, suggesting that the* σ *between the apo and SB state only differs by 20-30%. However, visual inspection of the traces in Figure 2 suggests a much more marked difference. Are the values in Figure 4 obtained from a distribution containing the distributions of FRET values for all molecules, or are the* σ*'s obtained at level of individual traces?*

The values for σ in Figure 5 are obtained by aggregating FRET values from all molecules.